# Stabilization of G-Quadruplexes Modulates the Expression of DNA Damage and Unfolded Protein Response Genes in Canine Lymphoma/Leukemia Cells

**DOI:** 10.3390/ijms26209928

**Published:** 2025-10-12

**Authors:** Beatriz Hernández-Suárez, David A. Gillespie, Ewa Dejnaka, Bożena Obmińska-Mrukowicz, Aleksandra Pawlak

**Affiliations:** 1Department of Pharmacology and Toxicology, Faculty of Veterinary Medicine, Wroclaw University of Environmental and Life Sciences, 50-375 Wroclaw, Poland; 2IRSD—Institut de Recherche en Santé Digestive, Université de Toulouse, INSERM, INRAE, ENVT, UPS, 31300 Toulouse, France; 3Instituto de Tecnologías Biomédicas, Facultad de Medicina, Campus Ciencias de la Salud, Universidad de La Laguna, 38071 La Laguna, Tenerife, Spain; 4Department of Physiology and Pharmacology, University of Georgia, Athens, GA 30602, USA; 5SMART Pharmacology, Precision One Health Initiative, University of Georgia, Athens, GA 30602, USA

**Keywords:** G4, DDR, UPR, PhenDC3, canine lymphoma/leukemia cell lines

## Abstract

G-quadruplexes have been identified as a promising anti-cancer target because of their ability to modulate the stability of mRNAs encoding oncogenes, tumor suppressor genes, and other potential therapeutic targets. Deregulation of DNA damage and Unfolded Protein Response pathways in cancer cells may create vulnerabilities that can be exploited therapeutically. Previous studies have shown variations in the relative expression of DDR and UPR components in canine lymphoma and leukemia cell lines CLBL-1, CLB70, and GL-1. In the present study, we report the presence of G-quadruplex structures in these canine cell lines. Downregulation of the expression of DDR and UPR components at the mRNA level was observed in the CLBL-1 and CLB70 cell lines after stabilization of G4 structures using the ligand PhenDC3. In contrast, in GL-1 cells, important components of the DDR pathway, such as PARP1, GADD45A, and PIK3CB were upregulated in response to PhenDC3 treatment. Downregulation of DDIT4 mRNA expression, which encodes an important UPR component, was detected in the CLBL-1 and GL-1 cell lines after PhenDC3 exposure. These results suggest that G4 structures can be used to manipulate the expression of potential targets to treat lymphoma in dogs. A substantial enrichment of DNA replication and pyrimidine metabolism pathways was found in the GL-1 cell line after G4 stabilization. This finding suggests that PhenDC3 may induce DNA replication stress in this cell line. Collectively, these results support the feasibility of employing canine cancer cells as a model system to investigate the role of G-quadruplex structures in cancer.

## 1. Introduction

G-quadruplexes (G4s) are non-canonical DNA or RNA structures formed by tetrads of guanines bonded by Hoogsteen hydrogen bridges. G4s can regulate gene expression by increasing mRNA stability, inducing alternative splicing, influencing transcription termination by controlling 3′-end processing, and by regulating the final step of mRNA translation [1,2,3] (Figure 1). Generally, G4s stop translation by preventing ribosome binding to the mRNA, but when G4s are located upstream the internal ribosome entry site (IRES), they act oppositely and enhance translation [4]. Specifically, G4 structures can modulate gene expression in cancer [5,6,7,8]. G4 structures can also interfere with DNA replication and repair, leading to increased DNA damage and replication stress [9].

The use of G4 ligands as a potential anti-cancer therapy is currently under investigation [10]. These compounds stabilize G4 structures and may exert anti-cancer effect either by inducing DNA damage selectively in cancer cells or by modulating the expression of oncogenes [11,12,13,14]. PhenDC3 is a well-characterized G4 ligand and stabilizer that has been widely employed to investigate the biological functions of G4 structures. It has been proposed as a therapeutic candidate, either as a monotherapy through the induction of DNA damage or in combination strategies exploiting synthetic lethality with inhibitors of other catalytic activities, such as WEE1 [15].

Currently, there is limited information available regarding G4 structures in canines. To our knowledge, only one relevant study has been reported, in which whole-transcriptome sequence profiles of two mast cell tumor models—one human and one canine cell line—were compared following exposure to the G4-binding anthraquinone derivative AQ1 [16]. It was found that in both cell lines, KIT- and MYC-related pathways were downregulated in the AQ1-treated group, while apoptosis was upregulated compared to the control group [16]. These results show that AQ1 treatment induces a similar transcriptional and biological response in human and canine cells, and that canines may be a viable translational model for studying the action of G4-binding compounds.

Lymphoma and leukemia are among the most common spontaneous tumors occurring in dogs. Unfortunately, although conventional chemotherapy often has a high initial response, many patients relapse [17]. Therefore, there is an urgent need for novel therapies for canine patients. The DNA damage response (DDR) and Unfolded Protein Response (UPR) are two important cellular stress pathways. In both humans and dogs, evidence suggests that alterations in the expression and function of the DDR and UPR pathways may be selected during tumorigenesis and contribute to the phenotype of the resulting cancer [18,19]. Previous studies have demonstrated that the expression levels of key components of the DDR and UPR, at both the protein and mRNA levels, vary in canine lymphoma and leukemia cells. These findings suggest that elements of these pathways may serve as potential therapeutic targets [20,21].

The present study aimed to evaluate the potential of G4s as therapeutic targets for the development of novel anti-cancer strategies by assessing their presence in canine lymphoma and leukemia cells and investigating how G4 stabilization influences the expression of DDR and UPR pathways.

## 2. Results

### 2.1. PhenDC3 Induces Cell Death in Canine Lymphoma and Leukemia Cell Lines

To understand cytotoxicity effects of PhenDC3, the half-maximal inhibitory concentration (IC50) was calculated. The control MDCK cell line (normal canine kidney cells) exhibited an IC50 value of approximately 12 µg/mL for PhenDC3, which was higher than that observed in the lymphoma and leukemia cell lines, where 50% inhibition of cell metabolic activity occurred at concentrations between 7 and 8 µg/mL (Table 1).

Next, cells were treated with DMSO or 5 µM of PhenDC3 for 24, 48, and 72 h to analyze patterns of observed cytotoxicity. The basal level of apoptosis in MDCK cells was relatively low, and PhenDC3 treatment induced little, if any, increase in the percentage of apoptotic cells compared to DMSO control at any time point (Figure 2A, blue). In the case of CLBL-1 and CLB70 cell lines, the pattern was very different. Here, around 25% of cells were apoptotic when treated with DMSO alone. The percentage of apoptotic cells increased to 30%, 50%, and 80% after PhenDC3 treatment for 24, 48, and 72 h, respectively, in both CLBL-1 and CLB70 cell lines (Figure 2A, orange and gray). The GL-1 cell line was more resistant to apoptosis compared to CLBL-1 and CLB70, showing an increase of up to 30% of apoptotic cells only after 72 h of treatment with PhenDC3 (Figure 2A, yellow). The percentage of necrotic cells showed greater variability among the four cell lines tested; for MDCK, only a few necrotic cells were detected (Figure 2B, blue). In CLBL-1, approximately 10% of cells were necrotic in both DMSO and PhenDC3 conditions after 24 h, increasing to 20% after 48 h in the cells treated with PhenDC3, and interestingly, decreasing to 15% after 72 h of PhenDC3 treatment (Figure 2B, orange). In CLB70, a slight increase to 10% of necrotic cells after PhenDC3 treatment for 48 and 72 h was noticed (Figure 2B, gray). Finally, whereas the GL-1 cell line was more resistant to PhenDC3-induced apoptosis, both the basal and induced levels of necrosis were substantially higher at 48 h than in the other cell lines (Figure 2B, yellow).

PhenDC3 is known to induce ssDNA lesions [22]. A marked increase in the γH2AX signal was observed following treatment with higher concentrations of PhenDC3 (10 or 30 μM) for 24 h (Appendix A). Treatment with 5 μM for 24 h resulted in a slight increase in γH2AX levels compared to untreated or control cells (Figure 3), although this difference did not reach statistical significance.

### 2.2. Confirmation of G4 Structures in Canine Lymphoma and Leukemia Cells

To validate that PhenDC3 binds to G4 structures in canine cells, the cells were incubated with the BioCyTASQ probe. In all four cell lines, an increased fluorescent streptavidin signal was detected in the PhenDC3-treated samples (Figure 4), indicating stabilization of G4 quadruplex structures.

### 2.3. G4 in DDR and UPR Targets

#### 2.3.1. Database Search to Explore RG4s on DDR and UPR Genes

Based on our previous publications, two DDR and three UPR genes were selected for the study. In human sequence, six RG4 structures were predicted in *CHEK1*, with eight predicted in *RAD51*, two of which have been experimentally validated in previous research studies (Table 2, DDR). For UPR genes, only predicted RG4 were found, one for *DDIT3*, two for *EIF2A*, and 15 for *ATF4* (Table 2, UPR). The sequences containing the RG4 identified in human showed matched alignments with the dog sequences in most of the cases, with the exception of three in *CHEK1*, two in *RAD51*, and seven in *ATF4* (Table 2, detailed sequence alignments on Appendix A).

#### 2.3.2. Changes in Canine Cells Observed at the mRNA Level in the DDR and UPR Pathways After Treatment with PhenDC3 at 5 μM for 24 h

In the non-cancerous MDCK cell line, treatment with PhenDC3 led to a significant downregulation of the expression of the DDR genes *CHEK1* and *RAD51* compared to DMSO-treated controls. Similarly, two UPR genes, *EIF2A* and *ATF4*, exhibited statistically significant reductions in expression following PhenDC3 treatment. Although the expression of a third UPR gene, *DDIT3*, also declined slightly, this change did not reach statistical significance (Figure 5, upper left quadrant). In contrast, the lymphoma/leukemia cell lines exhibited more variable responses, with GL-1 in particular showing notable differences in UPR gene expression changes. In the CLBL-1 cell line, expression of DDR genes was decreased when G4 structures were blocked with PhenDC3 compared to DMSO, although the scale of the changes in each case did not reach statistical significance. UPR gene expression also decreased in the samples with PhenDC3 compared to DMSO, but the changes were only statistically significant for *DDIT3* and *EIF2A* (Figure 5, upper right quadrant). In the case of CLB70, all tested gene expressions were significantly decreased in PhenDC3 compared to DMSO, except for the UPR gene *ATF4* (Figure 5, lower left quadrant). Finally, the GL-1 cell line exhibited the most divergent response among the cell lines analyzed. While a significant decrease in DDR gene expression was observed—consistent with the other cell lines—PhenDC3 treatment led to a significant upregulation of the UPR genes *EIF2A* and *ATF4*, in contrast to the downregulation seen in the other models (Figure 5, lower right quadrant).

### 2.4. RNA Sequencing Analysis

#### 2.4.1. Quantification Analysis

A table of quantitative results of gene expression can be found as an Excel file in the Appendix A—result folder, directory: Quant/Count/gene_count.xls (Supplementary Data access through the link provided in the Data Availability Statement).

A boxplot represents the distribution of gene expression between the replicates and different biological samples (Appendix A). A Pearson correlation matrix showed that the correlation of gene expression between the samples was higher than 0.9 for all the cases, meaning that the similarity between replicates was high, so the conditions of the experiments were optimal (Appendix A). Another quality analysis performed was a principal components analysis (PCA). The PCA was performed on the FPKM (Fragments Per Kilobase of transcript per Million mapped reads) value of all samples, and it showed that the replicates are grouping as expected, and the samples between different groups were widely separated. This analysis focused on comparing the samples from different conditions (DMSO or PhenDC3 treated) within the same cell line. No comparison between the samples from different cell lines was relevant (Appendix A).

#### 2.4.2. Differential Gene Expression Analysis

##### Differential Gene Expression Analysis for Each Comparison Across Cell Lines

All three cell lines presented the same pattern, with more downregulated than upregulated genes in the samples treated with PhenDC3 than in the DMSO group (Figure 6). In the CLBL-1 cell line, of the 978 genes expressed, 786 were downregulated and 192 upregulated. In CLB70, a total of 1229 genes were analyzed, of which 931 were downregulated and 298 were upregulated. In the GL-1 cell line, a total of 1431 genes were significantly expressed, 1031 were downregulated, and 400 were upregulated when comparing PhenDC3 treatment to DMSO.

##### Alterations in the Activity of the DDR and UPR Pathways Observed Across the Three Analyzed Cell Lines Following Treatment with the G4 Ligand

G4 blocking seemed to induce a general downregulation of gene expression in the three cell lines tested: CLBL-1 presented 80% of downregulated genes and 20% upregulated in the PhenDC3 condition compared to DMSO, CLB70 77% downregulated and 23% upregulated, and GL-1 72% downregulated and 28% upregulated (Figure 7). In the CLBL-1 cell line, 18 genes were related to the DDR pathway, of which 11 were downregulated and seven were upregulated. Nine genes were related to the UPR pathway, eight downregulated and one upregulated (Figure 7A). In the case of CLB70, 16 genes were related to the DDR pathway, 13 downregulated and only three upregulated, while among 14 genes related to the UPR, 10 were downregulated and four upregulated (Figure 7B). For the third cell line, GL-1, 24 genes were DDR-related, 14 downregulated and 10 upregulated; and of 15 UPR-related genes, 13 were downregulated and two upregulated (Figure 7C).

Three important DDR genes, *PARP1*, *PIK3CB*, and *GADD45A*, were downregulated in CLBL-1 and CLB70 after G4 blocking but upregulated in the GL-1 cell line (Table 3, in yellow). *PARP1* expression showed cell line–specific differences in response to PhenDC3 treatment. In the GL-1 cell line, *PARP1* exhibited a log2 fold change (log2FC) of 1.06 when comparing PhenDC3 to DMSO treatment, indicating an approximate two-fold increase in expression following G4 stabilization. In contrast, *PARP1* expression was significantly downregulated in CLBL-1 and CLB70 cells, with log2FC values of −1.91 and −1.11, respectively, corresponding to approximately four-fold and two-fold reductions in expression under PhenDC3 treatment compared to DMSO. *PIK3CB* gene showed a log2FC of −1.34 in CLBL-1, −1.08 in CLB70, and 1.03 in GL-1. Thus, *PIK3CB* was two times less expressed in PhenDC3 condition compared to DMSO in the CLBL-1 and CLB70 cell lines but two times more in GL-1. More significant changes occur in *GADD45A* gene, with values of log2FC of −4.98 in CLBL-1, −2.62 in CLB70, and 1.01 in GL-1. Thus, *GADD45A* was highly downregulated in CLBL-1 and CLB70 after blocking G4 structures with PhenDC3, while in GL-1, expression increased two-fold (Appendix A).

Among the UPR-related genes analyzed, *EIF4EBP1* was downregulated in both CLBL-1 and GL-1 cell lines following PhenDC3 treatment, whereas it was upregulated in CLB70 under the same conditions. The expression of *NCK2* gene was downregulated in CLBL-1 and CLB70 but upregulated in the GL-1 cell line. UPR gene *DDIT4* expression was downregulated in CLB70 and GL-1 but not significantly expressed in CLBL-1 (Table 3, in green). The expression of *EIF4EBP1* was three times lower in PhenDC3 treatment compared to DMSO in CLBL-1 and GL-1, with log2FC values of −1.5 and −1.57, respectively. The *NCK2* gene expression was slightly downregulated in CLBL-1, with a value of log2FC of −0.11. In CLB70, it was two times less expressed in PhenDC3 than in DMSO treatment, with a log2C value of −1.27. In GL-1, interestingly, it was two times higher expressed in PhenDC3 compared to DMSO, with a log2FC of 1.07. The expression of the *DDIT4* gene was three times lower in PhenDC3 treatment compared to DMSO in CLB70 and GL-1, with log2FC values −1.29 and −1.22, respectively (Appendix A).

#### 2.4.3. Enrichments Analysis

The complete results are provided in the Appendix A, accessible Via the link included in the Data Availability section.

##### Gene Ontology (GO) Analysis

GO analysis annotates genes to biological processes, molecular functions, and cellular components in a directed acyclic graph structure (Supplementary Data access through the link provided in Data Availability Statement).

For the CLBL-1 cell line, six processes were significantly upregulated in PhenDC3 compared to the DMSO condition. Those six processes were related to transmembrane signaling, including the G-protein-coupled receptors (GPCRs) signaling and activity (Table 4, CLBL-1). For the CLB70 cell line, 38 processes were significantly upregulated in PhenDC3 compared to the DMSO condition. From the 38 pathways, 16 were related to signaling pathways (including the GPCRs signaling and activity), 10 to structural and biological function, six related to DNA packaging, and six related to cell death pathways (Table 4, CLB70). For the GL-1 cell line, six processes were significantly downregulated and two significantly upregulated in PhenDC3 compared to the DMSO condition. The six upregulated were signaling pathways, including the GPCR activity, and the two downregulated were DNA replication and DNA metabolic processes (Table 4, GL-1).

##### Kyoto Encyclopedia of Genes and Genomes (KEGG)

KEGG analysis annotates genes to the pathway level. For the CLBL-1 cell line, six pathways were significantly downregulated in PhenDC3 compared to DMSO treatment. Five of those six pathways were related to different types of cancer (Table 5, CLBL-1). For the CLB70 cell line, eight pathways were significantly upregulated in PhenDC3 compared to the DMSO condition. From the eight pathways, two were related to diseases/medical conditions, and the other six to structural and biological functions of the cell (Table 5, CLB70). For the GL-1 cell line, 14 pathways were significantly altered in PhenDC3 compared to the DMSO condition. Five of the seven downregulated were signaling pathways, and the other two were related to alterations in response to therapies. Of particular interest were the upregulated genes, many of which were associated with DNA replication, DNA metabolic processes, and cancer-related pathways (Table 5, GL-1).

## 3. Discussion

### 3.1. PhenDC3 Stabilizes G4 and Induces Mild DNA Damage in Canine Lymphoma/Leukemia Cells

It is known that cancer cells can present higher numbers of G4 structures compared to non-cancerous cells [7]. One example includes a study in which non-neoplastic and liver-derived tumor samples from a patient were analyzed by immunohistochemistry, revealing increased G4 formation in the tumor tissue compared to the non-neoplastic counterpart [26].

In the present study, we demonstrated that canine lymphoma/leukemia cell lines exhibit greater sensitivity to PhenDC3 compared to the non-cancerous MDCK cell line (Table 1). Furthermore, AV/PI staining revealed that PhenDC3 predominantly induced apoptosis in CLBL-1 and CLB70 cells over time, whereas GL-1 cells primarily underwent necrosis. In contrast, MDCK cells appeared relatively resistant to both apoptotic and necrotic cell death under the concentrations tested (Figure 2). PhenDC3 induced mild DNA damage in canine cancer cell lines at a concentration of 5 µM after 24 h of treatment, whereas no detectable DNA damage was observed in the non-cancerous MDCK cells under the same conditions (Figure 3 and Appendix A). It has been reported that the location of G4 structures on DNA are highly associated with γH2AX sites, which may be linked to the fact that there is an absence of helicases for unwinding the G4 structures [9]. The γH2AX signal detected after treatment with PhenDC3 is the first evidence supporting the presence of G4s on the studied cell lines. Although G4 structures were not individually quantified in the present study, previous evidence supports that the PhenDC3 ligand stabilizes G4 structures in canine cells. In line with this, imaging analysis revealed significantly increased BioCyTASQ signal intensity in PhenDC3-treated samples compared to DMSO controls across all four cell lines tested (Figure 4), meaning that PhenDC3 has stabilized the G4 structures, as expected [27].

The BioCyTASQ probe results, together with the relative sensitivity of canine lymphoma/leukemia cells to PhenDC3 compared to non-transformed MDCK cells, suggest that stabilization of G4 structures exerts cancer-specific cytotoxic effects. The exact mechanism by which PhenDC3 binds to G-quadruplex structures and elicits its effects in each specific cell line remains to be fully elucidated.

### 3.2. DDR and UPR Components Are Downregulated or Upregulated Depending on the Cell Type After G4 Stabilization with PhenDC3

Utilizing QUADRatlas [28], we found that some genes have predicted RG4s on their sequences in humans (*CHEK1*, *RAD51*, *DDIT3*, *EIF2A*, and *ATF4)*, whereas in the case of *RAD51*, *PARP1*, *GADD45A*, and *DDIT4*, the presence of functional G4 structures in their sequences has been validated (Table 2 and Appendix A) [23]. Due to the similarities and percentage of homology between canine and human gene sequences, especially in highly conserved DDR and UPR members [20,21], both species likely share the G4 structures. A BLAST alignment comparing human and dog sequences confirmed that most of the RG4s detected in humans were conserved in the canine sequence (Appendix A). Further analyses are needed to verify the presence of G4 in those genes in canines.

One of the roles of G4s is to control mRNA translation into proteins [8]. A blocked G4 structure in the mRNA can cause reverse transcriptase stalling [23], impeding mRNA translation [29]. During RNA sequencing, blocked G4s in the RNA can cause the enzyme stalling, avoiding the conversion of RNA into cDNA. The results demonstrated that, in all three cell lines tested, a greater number of genes were downregulated than upregulated following G4 structure stabilization with the PhenDC3 ligand (Figure 6), including DDR- and UPR-related genes (Figure 7). This trend is consistent with findings from previous studies employing other G4-stabilizing ligands such as AQ1, CX-5461, and PDS [13,30].

Other G4 ligands have been shown to generate abnormalities in DDR-related gene expression. Some examples are the G4 ligand RHPS4, which induces a strong reduction in Chk1 and RAD51 proteins and transcript levels, and the triazine derivative 12459, which triggers a delay of ATR-Chk1 response and p53 activation in response to DNA damage [31,32]. Another study used two different G4 ligands and pooled the target genes that were depleted after the use of both drugs. The results showed that DDR genes *BRCA2*, *REV1*, *POLQ*, *ATM*, and *ATR*, were downregulated, accompanied by a decrease or depletion of mRNA expression after the treatment with both CX-5461 and pyridostatin (PDS) [30]. In our experiments, the stabilization of G4s was affecting *RAD51* and *CHK1* mRNA levels (Figure 5), suggesting that translation of those two genes could potentially be regulated directly or indirectly by G4 structures in the analyzed canine cells. Some preliminary analysis checking the protein expression levels by Western blot revealed that there are no changes for Chk1 in any of the cell lines comparing treated and non-treated (Appendix A). However, for the CLB70 cell line, analysis of Rad51 protein detected a significantly increased expression level after the treatment with PhenDC3 for 24 h (Appendix A). It could be possible that the G4s in RAD51 for the CLB70 cell line are formed near the IRES, enhancing IRES-mediated translation. Further research is needed to corroborate the effect of G4s on the expression of *RAD51* and *CHK1* in these cell lines, and to verify if the decrease in mRNA expression by stabilization of the G4s is affecting the translation of their encoded proteins.

UPR-related genes are also affected by the stabilization of G4s. The *DDIT3* mRNA was significantly decreased in cells CLBL-1 and CLB70, but the observed decrease was not significant in the cases of MDCK and GL-1 cell lines (Figure 5). According to the study of Micco et al., *DDIT3* is a gene that is downregulated under low concentrations of tetra-substituted naphthalene diimide derivatives, which are potent stabilizers of gene promoter DNA quadruplexes. It is possible that it reacts in the same way with PhenDC3 [33]. The *EIF2A* mRNA was significantly decreased when treated with PhenDC3 compared to DMSO for the MDCK, CLBL-1, and CLB70 cell lines, but significantly increased in the GL-1 cell line (Figure 5). A similar pattern was observed for the expression of *ATF4* mRNA. Previous studies have reported that *ATF4* mRNA expression is reduced following treatment with the G4 ligand PDS [34], which contrasts with the upregulation observed in the GL-1 cell line in the present study, but aligns with the downregulation seen in the other cell lines analyzed. The results observed in the GL-1 cell line are unexpected, as they contrast with the anticipated response; notably, PhenDC3 appears to induce the mRNA expression of *ATF4* and *EIF2A* in this cell line.

These results demonstrate that G4 stabilization in the three canine cancer cell lines we have studied affects the expression of multiple DDR and UPR genes in complex and different ways in each cell line. The next step will be to understand the mechanisms behind these phenomena, and if the changes are due to DNA damage induced by DBS formation for the treatment with the ligand or if these changes are a consequence of transcription inhibition by blocking the mRNAs.

### 3.3. GL-1 Might Be Suffering Oxidative Stress When Treated with PhenDC3

A noteworthy observation is that in the GL-1 cell line, KEGG analysis showed that DNA repair pathways (Base excision repair, Mismatch repair) are upregulated in this cell line by PhenDC3 treatment, but not in the other cell lines analyzed (Table 5). As shown in Figure 2B, the GL-1 cell line exhibited higher levels of necrosis compared to the other cell lines, both under basal conditions and following treatment with PhenDC3. Reactive oxygen species (ROS) are key mediators of necrosis [35], and it has been reported that ROS-induced damage to G4 structures can lead to increased expression of the *EIF2A* gene. [36]. Those observations suggest that oxidative stress may be a contributing factor to the distinct response observed in the GL-1 cell line. Fleming et al. reported that oxo-guanines located on G-rich DNA sequences that form G4 structures enhance gene expression by activating the base excision repair (BER) pathway [36]. It is known that one of the oxidative modifications by ROS is the formation of oxo-guanines [37,38]. PhenDC3-treated GL-1 cells suffer from ROS that could cause the formation of oxo-guanines. After treating the cells with PhenDC3, the oxo-guanine will form a tetrad with other guanines in the genes, which will be recognized and activate the BER. Further research is needed to corroborate this hypothesis.

### 3.4. The Possibility of Using PhenDC3 in Combinational Therapy

The use of G4 stabilizers in therapy has already been proven; a synthetic lethal effect was described by using the G4 stabilizing ligand CX-5461 (Pidnarulex) [39]. This drug is under clinical trials in phase I for humans—“Study of CX-5461 in Patients With Solid Tumours and BRCA1/2, PALB2 or Homologous Recombination Deficiency (HRD) Mutation” [40].

The use of G4-stabilizing compounds as a therapeutic strategy in canines remains largely unexplored. To determine whether PhenDC3 is a viable candidate for clinical application in canine cancer treatment, further comprehensive research is required. Based on the results obtained in this research, some future directions are proposed:-The upregulation of *PARP1* observed in GL-1 after blocking the G4s. Many available PARP inhibitors (PARPi) are used in clinics, but many are inducing resistance over time [41], and it has been shown that cancer cells use upregulation of PARP1 as a mechanism to become resistant to PARPi [42]. Treatment with olaparib has been tested in GL-1 and CLBL-1 by our group, showing that both cell lines are dying in a concentration and time-dependent manner, and in fact, GL-1 was less sensitive compared to CLBL-1 to olaparib treatment [43]. One potential avenue for future investigation is the combination of G4 ligands with PARPi in the canine cell lines studied, with GL-1 serving as a promising model for exploring therapeutic strategies in *PARP1*-upregulated cells.-*DDIT4* overexpression is related to poor prognosis in acute myeloid leukemia (AML) human patients treated only with chemotherapy [44]. The results of the presented study showed that using a G4 ligand reduces *DDIT4* expression in CLB70 and GL-1 cell lines, suggesting that the combination of PhenDC3 with the usual chemotherapy could improve the efficiency of the treatment and conditions of the patients. Given the physiological and molecular similarities between humans and canines, it is highly plausible that *DDIT4* expression patterns observed in human AML may also be present in canine patients.

## 4. Materials and Methods

### 4.1. Cell Material and Conditions

#### 4.1.1. Cells and Culture

In the presented study, the following canine cell lines were used: B-cell lymphoma cell line—CLBL-1, from Barbara Rütgen (Institute of Immunology, Department of Pathobiology, the University of Vienna, Vienna, Austria) [45], B-cell leukemia cell line—GL-1 from Yasuhito Fujino and Hajime Tsujimoto (Department of Veterinary Internal Medicine at the University of Tokyo, Tokyo, Japan) [46], and the B-cell chronic lymphocytic leukemia cell line—CLB70 [19], which was established with the participation of researchers from our laboratory. The MDCK was used as a non-cancerous control. The MDCK cell line was obtained from the ECACC.

CLBL-1 and GL-1 cell lines were cultured in RPMI 1640 culture media (Institute of Immunology and Experimental Therapy, Polish Academy of Science, Wrocław, Poland), while CLB70 cell line was cultured in Advanced RPMI (Gibco, Grand Island, NY, USA). MDCK cells were cultured in Dulbecco’s Modified Eagle’s Medium (DMEM) (Sigma Aldrich, Steinheim, Germany). The different culture media were supplemented with 2 mM L-glutamine (Sigma Aldrich, Steinheim, Germany), 100 U/mL of penicillin, 100 µg/mL of streptomycin (Sigma Aldrich, Steinheim, Germany), and 10–20% heat-inactivated fetal bovine serum (FBS) (Gibco, Grand Island, NY, USA). The growing conditions for the cells were 5% CO_2_ and 95% humidified air, at 37 °C in 25 cm^2^ cell culture flasks (Corning, New York, NY, USA). A monthly for mycoplasma contamination was performed on the cells and determined to be negative.

#### 4.1.2. Drugs and Treatments

The G4 ligand PhenDC3 (Sigma Aldrich, Steinheim, Germany) was diluted in DMSO and stored at −20 °C until use. Cells were treated with PhenDC3 in different concentrations for 24, 48, or 72 h (concentrations and time are specified for each experiment). DMSO alone was added to control cells.

### 4.2. Cell Proliferation Assay

The cell proliferation assay was performed using the MTT test (Sigma Aldrich, Steinheim, Germany). A total of 7 × 10^5^ (CLBL-1 and CLB70), 6 × 10^5^ (GL-1), or 1 × 10^6^ (MDCK) cells per mL were seeded in a 96-well-plate (Thermo Fisher Scientific, Roskilde, Denmark), and the PhenDC3 was added in increasing concentrations (0.78, 1.5, 3.125, 6.25, 12.5, and 25 µM). After incubation for 24 h, the MTT solution (5 mg/mL) was added to each well. Then, 24 h later, the optical density of wells was measured using a microplate reader (Spark, Tecan, Männedorf, Switzerland) at a reference wavelength of 570 nm. IC50 values represent the means of three independent experiments, each performed in triplicate.

### 4.3. Apoptosis Study

Cells were treated with 5 µM PhenDC3 and incubated for 24, 48, or 72 h, followed by staining with annexin V conjugated to fluorescein isothiocyanate (FITC) and PI. The cells were suspended in a binding buffer together with annexin V-FITC and PI (PI concentration 1 µg/mL) and incubated for 10 min at room temperature. Flow cytometry analysis was performed using a CytoFlex flow cytometer (Beckman Coulter, Inc., Headquarters Indianapolis, IN, USA), with CytExpert© software version 2.5.0.77 (Beckman Coulter, Inc.) being used for data analysis.

### 4.4. Western Blot

A total of 6 × 10^5^ cells/mL were cultured in 10 mL of medium in a 25 cm^2^ culture flask. Cells were treated with PhenDC3 at a concentration of 5, 10, and 30 μM for 24 h and directly lysed in urea/SDS buffer (composition as published in [21]). Using a BioRad Mini-PROTEAN Tetra Vertical Electrophoresis Cell system, 8–12% bis-tris acrylamide gels were prepared to run the samples. Using a BioRad Mini Trans-Blot^®^ Cell (Hercules, CA, USA) for wet transfer and a BioRad Trans-Blot^®^ Turbo™ Transfer System device (Hercules, CA, USA) for semi-dry transfer, the samples were transferred into nitrocellulose membranes for incubation with the antibodies.

The primary antibodies used were β-Actin C4 (sc-47778, from Santa Cruz Biotechnology Inc. (Dallas, TX, USA)) diluted 1:1000 in 3% milk in TBS-T, anti-gamma H2AX 9F3 (ab26350, from Abcam (Cambridge, UK)) diluted 1:1000 in 3% BSA in TBS-T, Chk1 G-4 (sc-8408, from Santa Cruz Biotechnology Inc. (Dallas, TX, USA)) diluted 1:1000 in 3% BSA in TBS-T, and Rad51 G-9 (sc-377467, from Santa Cruz Biotechnology Inc. (Dallas, TX, USA)) diluted 1:600 in 3% BSA in TBS-T. Goat Anti-Mouse Immunoglobulins/HRP (#P0447 at 1:20,000 concentration in TBS-T solution) was used as the secondary antibody, from Dako, now part of Agilent (Santa Clara, CA, USA). Two replicates were analyzed statistically, with the means and standard deviations represented in the graph. A *t*-test analysis was performed to determine the statistical difference between the DMSO control and PhenDC3-treated samples.

### 4.5. Immunofluorescence Microscopy—Biotrackers BioCyTASQ

A total of 0.5 × 10^6^ MDCK cells, 1 × 10^6^ GL-1 cells, and 1.5 × 10^6^ CLB70 and CLBL-1 cells were cultured in 6-well plates (Thermo Fisher Scientific, Roskilde, Denmark) and were treated with DMSO or PhenDC3 at a 5 µM concentration for 24 h. Cells were treated with the BioTracker™ BioCyTASQ G-quadruplex (G4) cell probe (Sigma Aldrich, Steinheim, Germany) at a concentration of 1 μM under dark conditions to enable the detection of G4s. After 24 h, cells were washed in PBS once and fixed in cold MeOH for 10 min. Cells were washed with PBS and subsequently incubated in a blocking solution of 1% BSA in PBS for 30 min at room temperature. To visualize BioCyTASQ under the microscope, cells were incubated with Molecular Probes™ Streptavidin–Cy3 (Sigma Aldrich, Steinheim, Germany) diluted 1:1500 in blocking solution for 1 h. Cells were washed with PBS before and after staining with DAPI (Sigma Aldrich, Steinheim, Germany) for 15 min. Cells were washed in PBS twice and then kept in PBS. For observation, 100 µL of cells were added to a well in a 96-well plate. When cells were at the bottom of the well, images were acquired using the inversed microscope Zeiss Axio Observer 7 (Jena, Germany) with a ×40 magnification. Image analysis was performed using FIJI software [47], the results are the mean of 3 independent experiments (5 measurements each condition). A *t*-test was performed comparing intensity measurements of the PhenDC3 condition normalized to DMSO to show statistical significance.

### 4.6. Database Search for G4 Structures on the Selected Genes

G-quadruplexes were expected to be found in canine genes of interest. A search using the database QUADRatlas [20] was made to check if the selected genes *CHEK1*, *RAD51*, *DDIT3*, *EIF2A*, and *ATF4* have RG4s (G4 in RNA) in the human sequence. Using BLAST^®^ services [24], the obtained human RG4s were compared to the dog sequence by analyzing the sequence alignment of the region where the RG4s are found in each gene. Based on the RNA sequencing results, a second search in the QUADRatlas database was performed looking for the RG4 structures on *PARP1*, *PIK3CB*, *GADD45A*, *EIF4EBP1*, *NCK2*, and *DDIT4* genes. The results were compared to dog sequence as it was performed for the previous genes.

### 4.7. qPCR

#### 4.7.1. RNA Isolation and Reverse Transcription

A total of 1 × 10^7^ CLBL-1, CLB70, and GL-1 cells cultured in 10 mL of media were pellet at 300 g at 4 °C and resuspended in 500 µL of TRIzol reagent (Invitrogen, Carlsbad, USA) and stored in Eppendorf tubes at −80 °C for further analysis. Total RNA Zol-Out™ D (A&A Biotechnology, Gdańsk, Poland) was used for the RNA isolation following the protocol provided in the isolation kit. Then, a reverse transcription was performed using The TranScriba noGenome Kit (A&A Biotechnology, Gdańsk, Poland), according to the manufacturer’s recommendations in the MJ Research PTC-100 thermocycler (Marshall Scientific, Hercules, CA, USA). cDNA products were stored at −20 °C until use.

#### 4.7.2. Primers Design

The *Canis lupus familiaris* nucleotide accession number sequences for mRNA of the target genes (TGs), *CHEK1*, *RAD51*, *CHOP*, *EIF2A*, *ATF4*, and two housekeeping genes (HKGs) *ACTB* and *RPLP0* were taken from the Ensembl database [48]. The sequences were transferred into the Primer 3 free software, selecting a size from 70 to 150 bp. Verification of specificity of the designed primers and their amplified sequences were performed by using the Nucleotide Basic Local Alignment Search Tool—Primer-BLAST (NCBI, Bethesda, DC, USA). Primer sequences were tested for primer dimers on the ThermoFisher website. Gene names, primer sequences for TGs and HKGs, amplicon size, as well as their respective gene accession numbers are summarized in Table 6.

#### 4.7.3. Gene Expression Analysis Using Real-Time PCR

Three independent experiments in duplicates were performed by real-time PCR. The reaction mix was prepared by mixing 5 μL of RT PCR Mix SYBR^®^ (A&A Biotechnology, Gdańsk, Poland), 0.4 μM of forward and reverse primers (Eurofins Genomics AT GmbH, Poland), and 2.5 μL of cDNA diluted in water (16.65 ng cDNA) per well. Real-time PCR was performed using a BioRad CFX instrument (Hercules, CA, USA) under the following conditions: pre-incubation at 95 °C for 2 min; 45 cycles of amplification: 15 s at 95 °C for denaturation, 30 s at 61.5 °C for annealing, and 15 s at 72 °C for elongation. Melting curve analysis was performed from 65 °C to 95 °C with 0.2 °C increments, followed by a final cooling step at 4 °C for 10 min to terminate the reaction. Gene detection analyses and primer specificity were further validated by melting curve analysis.

#### 4.7.4. Statistical Analysis of qPCR

Mean and standard deviation were calculated from the duplicates of the three independent experiments. A *t*-test analysis was performed to check for significant differences between the mRNA expression level among the different conditions (PhenDC3 to DMSO) or to identify significant differences between lymphoma/leukemia cell lines and the non-cancerous MDCK cells after PhenDC3 treatment.

### 4.8. RNA-Sequencing

RNA was obtained from CLBL-1, CLB70, and GL-1 cultures treated with DMSO or PhenDC3 at 5 µM concentration for 24 h and sequenced by Novogene (UK). Quality control of the samples was performed prior to sequencing. RNA quantity and absorbance ratios were measured to assess RNA purity (Appendix A). The abundance of transcripts reflects the expression level of the genes. Read counts are proportional to gene expression level, gene length, and sequencing depth. FPKM is a widely used metric for estimating gene expression levels, as it accounts for both sequencing depth and gene length, thereby normalizing fragment counts to enable more accurate comparisons across genes and samples. The differential gene expression analysis took the |log2(FoldChange)| ≥ 1 and padj ≤ 0.05 threshold to select genes that were significantly expressed in the analyzed samples. The two experimental conditions were compared to the observed differences in gene expression levels for each cell line.

To facilitate a detailed analysis of the DDR and UPR pathways, FPKM values were employed to quantify relative gene expression levels. A threshold in FPKM > 0.5 based on gene expression FPKM interpretation was used: expression level is below cutoff (0.5 FPKM), expression level is low (between 0.5 and 10 FPKM), expression level is medium (between 11 and 1000 FPKM), and expression level is high (more than 1000 FPKM) [49]. The DDR GO lists used for the intersection analysis were the same as presented in Table 1 in our previous publication [21], and the UPR-related GO lists were the same as presented in Table 1 in our previous publication [20]. The gene set information was obtained from the GSEA database [50,51]. Venn diagrams were prepared using Venny 2.0 [25].

## 5. Conclusions

In this study, we demonstrated that PhenDC3 effectively stabilizes G4 structures in canine lymphoma and leukemia cells. A 24 h G4 stabilization period was sufficient to induce notable changes in mRNA expression across a wide range of genes. Overall, a trend toward gene downregulation was observed, particularly among DDR and UPR genes in all three cell lines analyzed. Interestingly, the GL-1 cell line displayed an opposite pattern, with increased expression of several UPR components, suggesting a distinct cellular response to G4 stabilization. Additionally, PhenDC3 appeared to activate different cell death pathways depending on the cell line: apoptosis predominated in most cell lines, whereas in GL-1, PhenDC3 induced necrosis.

DDIT4 was found to be downregulated in both the CLB70 and GL-1 cell lines, indicating its potential as a combinatorial target with chemotherapy in dogs.

These findings highlight the utility of canine lymphoma and leukemia cell lines as valuable models for studying G4 biology and evaluating the therapeutic potential of G4-stabilizing ligands in cancer.

## Figures and Tables

**Figure 1 ijms-26-09928-f001:**
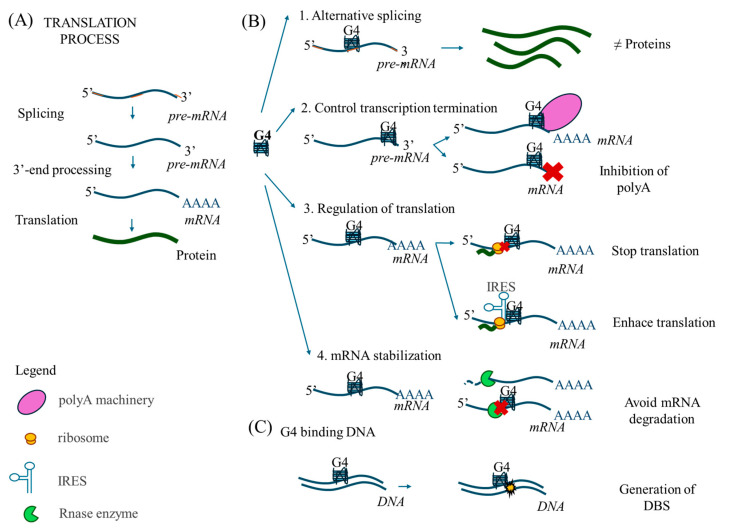
Roles of G-quadruplexes. (**A**) Scheme of translation. Pre-RNA, consisting of introns (orange) and exons (blue), introns are excised by splicing, leaving only the coding sections (exons). Next, the poly-adenine tail is added to the 3′ terminus to form mRNA, which is finally translated into protein (green curved-line). (**B**) G4 structures can regulate different steps of the translation process. 1—G4s can promote alternative splicing, resulting in the generation of different protein isoforms from the same mRNA. 2—G4s can regulate transcription termination by helping to recruit the polyadenylation machinery (polyA) or by inhibiting polyadenylation activity. 3—G4s can also regulate translation. G4s can prevent ribosomes from reading mRNA sequences and consequently stopping translation, but if G4s are located near the IRES, it will enhance translation. 4—G4 structures can induce mRNA stabilization by impeding degradation by endonucleases (red X symbolize how G4 blocks the endonuclease action). (**C**) G4 stabilization on DNA can lead to the formation of double-strand breaks (DSBs).

**Figure 2 ijms-26-09928-f002:**
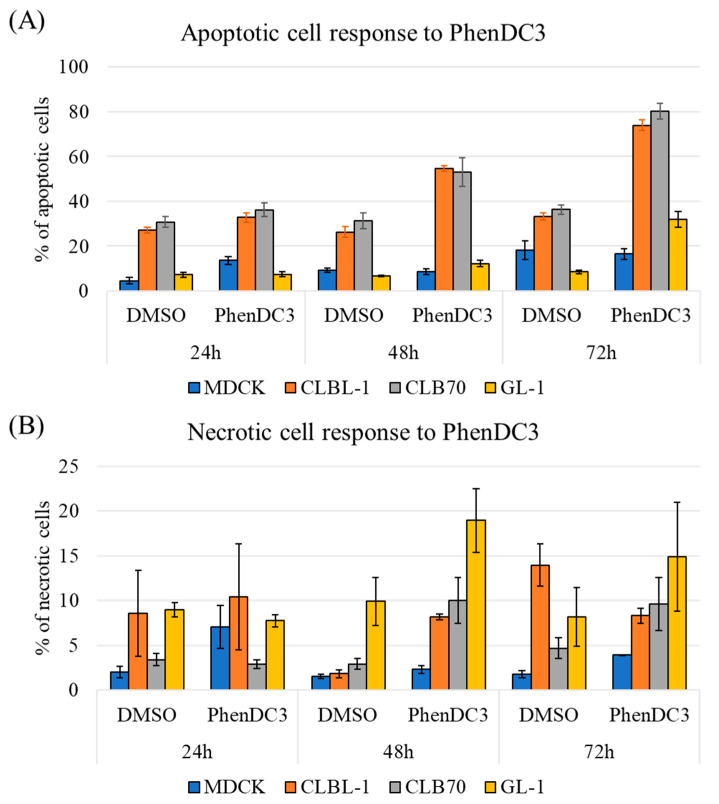
Percentage of apoptotic (**A**) and necrotic (**B**) cells was estimated by flow cytometry using Annexin V and propidium iodide (PI) staining. Four cell lines were used, MDCK as an example of a non-cancerous canine cell line, and three lymphoma and leukemia canine cell lines. Cells were treated with DMSO as a control or 5 µM PhenDC3 for 24, 48, and 72 h. A total of 20,000 events counted for each condition and replicate. Results of three independent experiments.

**Figure 3 ijms-26-09928-f003:**
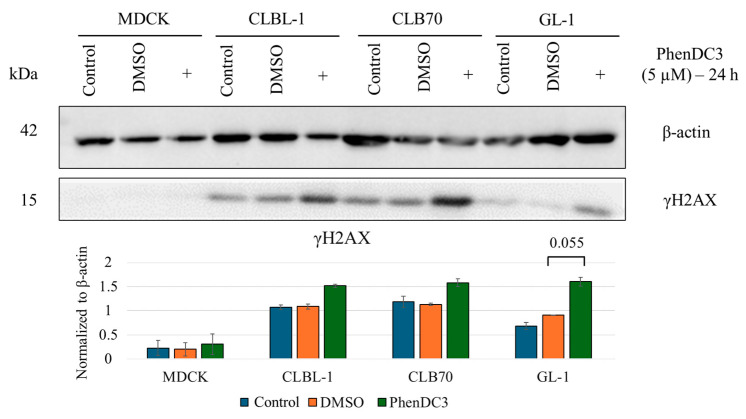
PhenDC3 induces DNA damage in cancer cell lines at a concentration of 5 μM. Western blot analysis of γH2AX was performed on four cell lines under three conditions: untreated control (Control), DMSO-treated (DMSO), and treated with the G4 ligand PhenDC3 (+). The mean of the normalized γH2AX signal quantification is presented with error bars representing the standard deviation. A *t*-test revealed that the increase in γH2AX observed in GL-1 cells following PhenDC3 treatment was statistically significant. The data represent the results of two independent experiments.

**Figure 4 ijms-26-09928-f004:**
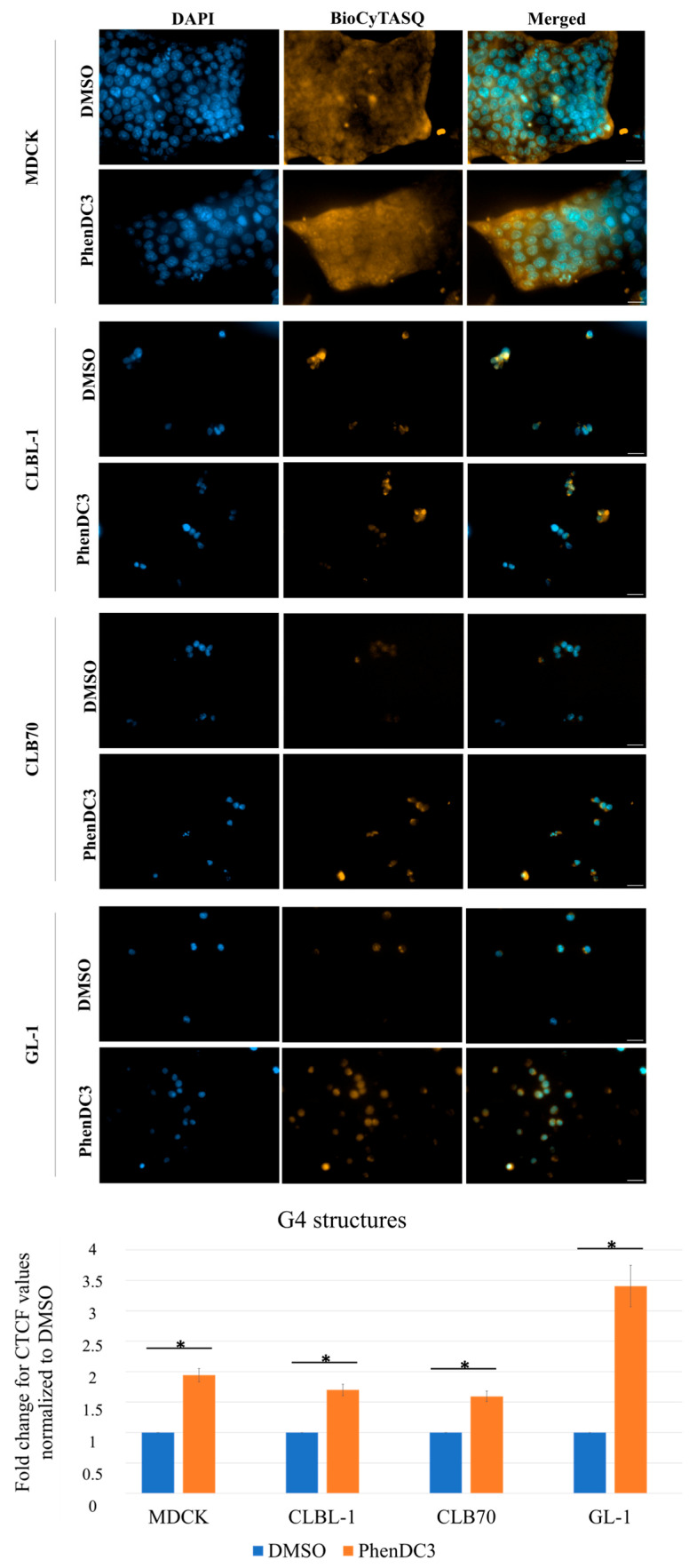
The presence of G4 structures was confirmed by BioCyTASQ-streptavidin staining in the four cell lines. Cells were treated with PhenDC3 at 5 µM for 24 h. Three independent experiments, five cells of each condition were measured for each experiment. Images acquired with x40 magnification. The scale is 20 µm. Corrected total cell fluorescence (CTCF) was measured using FIJI. The fold change was calculated by normalizing PhenDC3 to the DMSO condition. A *t*-test comparing the intensity of BioCyTASQ signal observed in PhenDC3-treated cells compared to DMSO alone was performed. *p*-values ≤ 0.05 represented with an asterisk (*).

**Figure 5 ijms-26-09928-f005:**
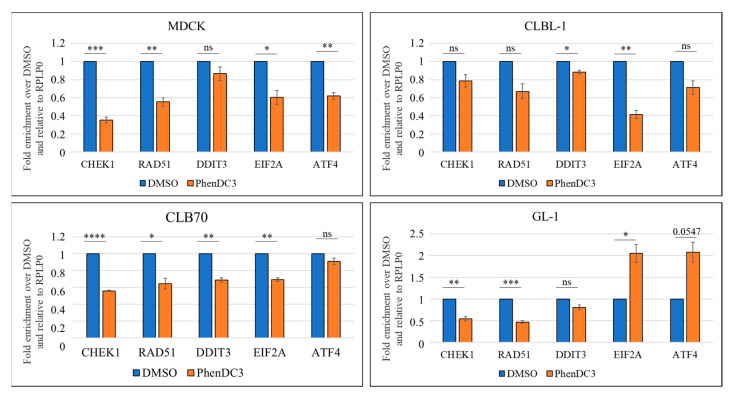
Fold change enrichment of the mRNA expression levels comparing DMSO to 5 µM PhenDC3 treatments in all tested cell lines. qPCR results showed that expression of DDR genes *CHEK1* and *RAD51* decreased after treatment with PhenDC3 when compared to DMSO. The UPR genes showed more variable results. The expression of the *DDIT3* gene tends to decrease in the four cell lines tested, but *EIF2A* and *ATF4* decreased in MDCK, CLBL-1, and CLB70, but increased in the GL-1 cell line. Results from three independent experiments. Significant *p*-values are represented with asterisk: ≤0.05 (*), ≤0.01 (**), ≤0.001 (***), ≤0.0001 (****). Non-significant *p*-values > 0.05 are represented as ns.

**Figure 6 ijms-26-09928-f006:**
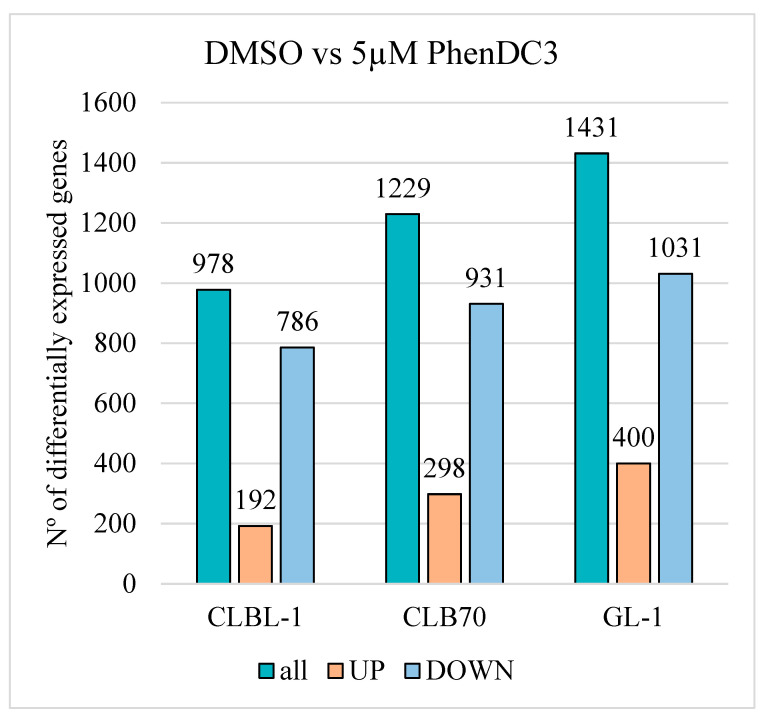
Graphical representation of differential gene expressions across cell lines. A graph illustrates the changes in gene expression between control (DMSO-treated) and PhenDC3-treated cells in three different cell lines. Comparisons were made between treatment conditions to assess the relative expression levels of genes. Overall, differential expression analysis revealed a greater number of downregulated genes than upregulated ones following PhenDC3 treatment in all three cell lines. Results from three independent replicates.

**Figure 7 ijms-26-09928-f007:**
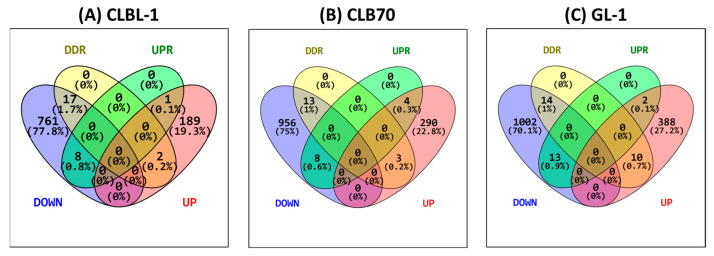
Venn diagrams representing the percentage of down- (blue) and up- (red) regulated genes of DDR- (yellow) and UPR- (green) related genes in the three different cell lines (**A**) CLBL-1, (**B**) CLB70, and (**C**) GL-1. Venn diagrams were prepared using the Venny tool [25]. Results from three replicates from independent experiments.

**Table 1 ijms-26-09928-t001:** Calculation of the IC50 of PhenDC3 by MTT test. The four different cell lines were treated with concentrations ranging from 0 to 25 µM and incubated for 24 h. The percentage of cell metabolic activity was measured, and the concentration inducing 50% inhibition was estimated.

	IC50 (µM)
MDCK	11.90 ± 0.13
CLBL-1	8.65 ± 1.63
CLB70	7.72 ± 0.64
GL-1	7.04 ± 0.80

**Table 2 ijms-26-09928-t002:** Number of RG4s predicted and experimentally described in DDR and UPR human genes, and the number of matched alignment sequences with the dog.

		Predicted *	Experimentally **	Reference	Alignment with the Dog Sequence	
DDR	*CHEK1*	6	-		3	-
	*RAD51*	8	2	[23]	6	2
UPR	*DDIT3*	1	-		1	-
	*EIF2A*	2	-		2	-
	*ATF4*	15	-		6	-

* Predicted RG4s are de novo predictions of RG4s on user-input sequences through three different algorithms. ** Experimentally, RG4s were determined by RG4-Seq, as found in citations. Comparison with the dog sequences using BLAST^®^ services version BLAST+ 2.17.0 [24].

**Table 3 ijms-26-09928-t003:** Differential expressions of DDR- and UPR-related genes following 24 h PhenDC3 treatment in three distinct cell lines. Gene expression analysis revealed both upregulation and downregulation of DDR and UPR genes across the GL-1, CLBL-1, and CLB70 cell lines after 24 h of PhenDC3 exposure.

	Gene	CLBL-1	CLB70	GL-1
DDR	*PARP1*	↓	↓	↑
	*PIK3CB*	↓	↓	↑
	*GADD45A*	↓	↓	↑
UPR	*EIF4EBP1*	↓	↑	↓
	*NCK2*	↓	↓	↑
	*DDIT4*	-	↓	↓

Legend: ↓ downregulated, ↑ upregulated, - not significantly expressed.

**Table 4 ijms-26-09928-t004:** Statistically significant enriched GO term list of DMSO-PhenDC3 comparison for the three different cell lines analyzed (padj > 0.05).

Cell Line	PhenDC3 to DMSO	GO
CLBL-1	DOWN	-
	UP	**GPCRs signaling pathway, GPCR activity**, transmembrane signaling receptor activity, signaling receptor activity, molecular transducer activity, signal transducer activity
CLB70	DOWN	-
	UP	SIGNALING (16/38): **GPCR signaling pathway**, regulation of signal transduction, regulation of cell communication, regulation of signaling, Ras protein signal transduction, regulation of Ras protein signal transduction, regulation of small GTPase mediated signal transduction, Rho protein signal transduction, regulation of Rho protein signal transduction, small GTPase mediated signal transduction, **GPCR activity**, signaling receptor activity, transmembrane signaling receptor activity, signal transducer activity, regulation of intracellular signal transduction, regulation of response to stimulus STRUCTURAL AND BIOLOGY FUNCTION (10/38): molecular transducer activity, GTPase binding, protein heterodimerization activity, guanyl-nucleotide exchange factor activity, Ras GTPase binding, small GTPase binding, enzyme binding, Ras guanyl-nucleotide exchange factor activity, Rho guanyl-nucleotide exchange factor activity, Rho GTPase binding DNA PACKING (6/38): Nucleosome, protein-DNA complex, DNA packaging complex, Chromatin, chromosomal part, chromosome CELL DEATH (6/38): apoptotic process, cell death, programmed cell death, regulation of cell death, regulation of apoptotic process, regulation of programmed cell death
GL-1	DOWN	ion channel activity, substrate-specific channel activity, channel activity, passive transmembrane transporter activity, **GPCR activity**, cation channel activity
	UP	**DNA replication**, DNA metabolic process

**Table 5 ijms-26-09928-t005:** Statistically significant enriched KEGG list of DMSO-PhenDC3 comparison for the three different cell lines analyzed (padj > 0.05). Relevant KEGG list related to cancer and/or DDR highlighted in bold.

Cell Line	PhenDC3 to DMSO	KEGG
CLBL-1	DOWN	CANCER TYPES (5/6): **Breast cancer, Colorectal cancer, Renal cell carcinoma, Gastric cancer, Endometrial cancer** OTHER (1/6): Glutamatergic synapse
	UP	-
CLB70	DOWN	-
	UP	DISEASE OR MEDICAL CONDITION (3/8): Amoebiasis, Alcoholism, Systemic lupus erythematosus STRUCTURAL AND BIOLOGICAL FUNCTION (5/8): Neutrophil extracellular trap formation, Cytokine-cytokine receptor interaction, Cell adhesion molecules, Toll-like receptor signaling pathway, Taurine and hypotaurine metabolism
GL-1	DOWN	SIGNALING PATHWAYS (5/7): Neuroactive ligand-receptor interaction, Aldosterone-regulated sodium reabsorption, Chemokine signaling pathway, Axon guidance, Inflammatory mediator regulation of TRP channels ALTERED RESPONSE TO THERAPIES (2/7): Morphine addiction, Endocrine resistance
	UP	DNA METABOLISM (2/7): Pyrimidine metabolism, Nucleotide metabolism DNA REPLICATION AND REPAIR (4/7): Cell cycle, **Base excision repair, DNA replication, Mismatch repair** CANCER TYPE (1/7): **Bladder cancer**

**Table 6 ijms-26-09928-t006:** Primer sequence information.

Gene Name	Primers 5′–3′ *	Amplicon Size	Gene Accession Number
** *CHEK1* **	F: TGGTTGACTTTCGGCTCTCT R: AAACCTTCTGGCTGCTCACA	102	XM_038503854.1
** *RAD51* **	F: TGTGGAGGCTGTTGCCTATG R: ATCGCCTTTGGTGGAACTCA	146	NM_001003043.1
** *DITT3* **	F: AGCCCTCACTCTCCAGATTC R: GCCACTCTGTTTCCGTTTCC	93	XM_022424097.2
** *EIF2A* **	F: TGCCTTGAATTCTCGCCAAA R: GTATCCCAGCTGTGCCATCT	82	XM_038432666.1
** *ATF4* **	F: GCTGGCTTTGGATGGGTTG7 R: CCAATCTGTCCCGGAGAAGG	70	XM_038679806.1
** *CLASPN* **	F: TGGAATCGATAAGGGCAGCT R: TGCCTTTGGATAGCTCAGTCT	114	XM_038687566.1
** *ACTB* **	F: ACGGGCAGGTCATCACTATT R: GGTAGTTTCATGGATGCCGC	104	NM_01195845.3
** *RPLP0* **	F: AGGGCATCTGGAGAACAACC R: TGAATACAAAACCCACATTCCCC	74	XM_038436105.1

* The letter “F” denotes the forward primer and “R” the reverse primer.

## Data Availability

All relevant data is contained within the article: The original contributions presented in the study are included in the article, in Appendix A and the RNA sequencing data can be found at https://data.mendeley.com/datasets/bwt93r942s/1 (accessed on 22 August 2025) further inquiries can be directed to the corresponding author.

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
