# Peer review of "Stabilization of G-Quadruplexes Modulates the Expression of DNA Damage and Unfolded Protein Response Genes in Canine Lymphoma/Leukemia Cells"

_ijms, 2025, doi:10.3390/ijms26209928_

Round 1

Reviewer 1 Report

Comments and Suggestions for Authors

The manuscript entitled "Stabilization of G-quadruplexes modulates the expression of 2 DNA damage and Unfolded Protein Response genes in canine 3 lymphoma/leukemia cells" by Suárez et al. showed G4 stabilization in the three canine cancer cell lines that affects the expression of DDR and UPR genes in cell line. Overall, the goal of the various experiments is introduced properly in the different sections, and if confirmed, the results obtained might deepen previous studies. However, the experimental designs and results interpretation must be improved.

  • How is the protein expression of  γH2AX connected to G4 structures? is γH2AX also regulated transcriptionally?
  • The microscope images don't have the scale bars, also in the figure legends the authors have not mentioned at what focus the images were taken at. Also why is the DAPI stain look blurry? 
  • It is advisable that the authors write the legends explaining details about the experiments and how many number (biological repeats) were the experiments carried performed throughout the manuscript. 

Author Response

For research article

Response to Reviewer 1 Comments

1. Summary

We would like to thank you for agreeing to review this paper and for the suggestions and comments that for sure will improve our paper. We hope we have addressed the various questions and queries in the revised manuscript. Here, we describe the changes that we have made regarding the comments, changes are highlighted in yellow. Please find detailed responses below.

2. Questions for General Evaluation

Reviewer’s Evaluation

Response and Revisions

Does the introduction provide sufficient background and include all relevant references?

Yes

Are all figures and tables clear and well-presented?

Must be improved

Is the research design appropriate?

Yes

Are the methods adequately described?

Yes

Are the results clearly presented?

Yes

Are the conclusions supported by the results?

Yes

3. Point-by-point response to Comments and Suggestions for Authors

Comments 1: How is the protein expression of γH2AX connected to G4 structures? is γH2AX also regulated transcriptionally?

Response 1: We appreciate the very interesting question. As we have mention in the introduction, G4 stabilization block replication inducing replication stress and DNA breaks. When DNA breaks, H2AX histone recognizes the damage site and serves as a docking point for the DNA repair components. Moreover, it has been reported that the location of G4 structures on DNA are highly associated with γH2AX sites, which may be linked to the fact that there is an absence of helicases for unwinding the G structures. We apologize for not having clear in the text, a better reasoning about connection of G4 and γH2AX is added in the discussion. Changes highlighted in yellow.

Comments 2: The microscope images don't have the scale bars, also in the figure legends the authors have not mentioned at what focus the images were taken at. Also why is the DAPI stain look blurry?

Response 2: We highly appreciate the comment as we consider that it is a necessary information that we missed. Information about the objective is added to the legend on the figure. Scales have been added to the merged images. Regarding DAPI staining, we have adjusted the colour balance and brightness to improve quality of images and reduce the blurriness. However, DAPI signal is still blurry in some images since cells were observed directly on a plate. Cells were kept in PBS after staining. For observation, 100uL of Lymphoma/leukemia cells were added to a well in a 96 well-plate. Later, when cells were at the bottom of the well, images were acquired. As cells were not on a slide, it was harder to get a better focus. Information about procedure is now added to the material and methods section. Changes highlighted in yellow.

Comments 3: It is advisable that the authors write the legends explaining details about the experiments and how manynumber (biological repeats) were the experiments carried performed throughout the manuscript. 

Response 3: We would like to appreciate the advice. We have included the information when missing Changes highlighted in yellow.

4. Response to Comments on the Quality of English Language

Point 1: The English is fine and does not require any improvement

Response 1: NA

5. Additional clarifications

There is nothing else to add.

Reviewer 2 Report

Comments and Suggestions for Authors

The authors of the manuscript investigate the effects of the G-quadruplex stabilizing ligand PhenDC3 on canine lymphoma and leukemia cell lines. The study demonstrates that PhenDC3 effectively stabilizes G4 structures in these cancer cells, leading to significant changes in gene expression, particularly downregulation of DNA damage response and unfolded protein response (UPR) pathway genes. Notably, apoptosis was induced in most cell lines, whereas the GL-1 cell line predominantly underwent necrosis and showed an opposite gene expression pattern for some UPR components. The research also highlights potential therapeutic implications, such as combining PhenDC3 with chemotherapy or PARP inhibitors, especially given the downregulation of genes like DDIT4. Overall, the findings support the use of canine lymphoma and leukemia cell lines as valuable models to study G4 biology and its therapeutic targeting in cancer. The manuscript is clearly written and well-illustrated; however, due to the mixing of results with detailed descriptions of the experimental methodology, it is not always easy to follow. My main concern regarding the manuscript is this blending of results and methods, which should be addressed. I also have a few other remarks that the authors should consider when preparing the final version of their paper. 

(1) As a rule, abbreviations should not be explained in the abstract. Instead, they should be introduced and defined upon their first occurrence in the main text of the manuscript. Here, some abbreviations are explained multiple times throughout the article, which should be avoided.

(2) The font size in Figure 1 is too small. Please consult the journal’s instructions for authors, which specify the minimum font size required for figures. The text should be large enough to remain fully legible when the article is printed in A4 format.

(3) In the Introduction, it would be valuable to include a few references, for example, related to G4 structures: Zok et al. ONQUADRO: a database of experimentally determined quadruplex structures. Nucleic Acids Research 2022 (doi:10.1093/nar/gkab1118). In the sentence "The use of G4 ligands as a potential anti-cancer therapy is currently under investigation." it would be appropriate to include a reference to recent work on this topic, for example: Iachettini et al. Therapeutic Use of G4-Ligands in Cancer: State-of-the-Art and Future Perspectives. Pharmaceuticals 2024 (doi: 10.3390/ph17060771).

(4) The Results section contains content that would be more appropriate for the Methods section (for example, most of section 2.1 should be moved to Methods, as it does not explain the results obtained in the analysis). The Results should present only the outcomes of the study, without detailed descriptions of experimental procedures, which belong in the Methods section.

(5) The authors discuss observations related to the stabilization of quadruplex structures. It would be interesting to know whether there are any insights or hypotheses regarding the actual topology of these structures—for example, the size of the quadruplexes (number of tetrads), whether they are unimolecular or multimolecular, and any other relevant structural details.

(6) The references section requires careful revision. In several places, the formatting and spelling of references are inconsistent or incorrect. For example, some entries contain artifacts (e.g., ref. 40), and in certain cases, fragments of titles or authors’ names are written in all capital letters (e.g., refs. 4 and 43). Moreover, some references are incomplete and lack essential bibliographic information (e.g., refs. 37 and 40).

Author Response

For research article

Response to Reviewer 2 Comments

1. Summary

We would like to thank you for agreeing to review this paper and for the suggestions and comments that for sure will improve our paper. We hope we have addressed the various questions and queries in the revised manuscript. Here, we describe the changes that we have made regarding the comments, changes are highlighted in pink. Please find detailed responses below.

2. Questions for General Evaluation

Reviewer’s Evaluation

Response and Revisions

Does the introduction provide sufficient background and include all relevant references?

Can be improved

Are all figures and tables clear and well-presented?

Can be improved

Is the research design appropriate?

Yes

Are the methods adequately described?

Yes

Are the results clearly presented?

Can be improved

Are the conclusions supported by the results?

Yes

3. Point-by-point response to Comments and Suggestions for Authors

Comments 1: As a rule, abbreviations should not be explained in the abstract. Instead, they should be introduced and defined upon their first occurrence in the main text of the manuscript. Here, some abbreviations are explained multiple times throughout the article, which should be avoided

Response 1: We appreciate the comment, and we have implemented the changes accordingly.

Comments 2: The font size in Figure 1 is too small. Please consult the journal’s instructions for authors, which specify the minimum font size required for figures. The text should be large enough to remain fully legible when the article is printed in A4 format

Response 2: We apologize for the small font on figure 1. Changes have been made to make it readable in a A4 format.

Comments 3: In the Introduction, it would be valuable to include a few references, for example, related to G4 structures: Zok et al. ONQUADRO: a database of experimentally determined quadruplex structures. Nucleic Acids Research 2022 (doi:10.1093/nar/gkab1118). In the sentence "The use of G4 ligands as a potential anti-cancer therapy is currently under investigation." it would be appropriate to include a reference to recent work on this topic, for example: Iachettini et al. Therapeutic Use of G4-Ligands in Cancer: State-of-the-Art and Future Perspectives. Pharmaceuticals 2024 (doi: 10.3390/ph17060771).. 

Response 3: The two publications suggested are both very interesting and useful. Unfortunately, we have decided to recline the proposal of including the first publication (Zok et al. ONQUADRO: a database of experimentally determined quadruplex structures. Nucleic Acids Research 2022 (doi:10.1093/nar/gkab1118)), as we have used another database in our analysis, and questions might be arisen if we cite a different database that we did not use. About the second publication suggested, we have decided to include it as suggested in the introduction. Changes are highlighted in pink.

Comments 4: The Results section contains content that would be more appropriate for the Methods section (for example, most of section 2.1 should be moved to Methods, as it does not explain the results obtained in the analysis). The Results should present only the outcomes of the study, without detailed descriptions of experimental procedures, which belong in the Methods section

Response 4: We appreciate the comment, and we have modified the text accordingly, the sentences that referred to methodology were either removed, rewrite or moved to methodology section. Changes are highlighted in pink.

Comments 5: The authors discuss observations related to the stabilization of quadruplex structures. It would be interesting to know whether there are any insights or hypotheses regarding the actual topology of these structures—for example, the size of the quadruplexes (number of tetrads), whether they are unimolecular or multimolecular, and any other relevant structural details

Response 5: We agreed that information about the topology of the G4 structures will enrich the publication. However, the scope of our study is to highly the importance of G4s to the veterinary research community, and to promote the use of canine cancer cells as a model to study G4. We also would like to remark that in our team we do not have the expertise neither the tools for such experiments. We apologize for not being able to proceed with this request.

Comments 6: The references section requires careful revision. In several places, the formatting and spelling of references are inconsistent or incorrect. For example, some entries contain artifacts (e.g., ref. 40), and in certain cases, fragments of titles or authors’ names are written in all capital letters (e.g., refs. 4 and 43). Moreover, some references are incomplete and lack essential bibliographic information (e.g., refs. 37 and 40)

Response 4: We highly appreciate the warning about the references. We have checked and correct all the references when needed.

4. Response to Comments on the Quality of English Language

Point 1: The English is fine and does not require any improvement

Response 1: NA

5. Additional clarifications

There is nothing else to add.

Reviewer 3 Report

Comments and Suggestions for Authors

The authors identified that G-4-quadruplexes regulate Transcriptional activity and DNA repair mechanisms in Lymphoma or leukemia cells. It is interesting to note that these B-cell lymphoma or leukemia lines behave differently with the stabilization of G4. Notably, the Upregulation of PARP1 in the GL-1 cell line has a therapeutic value.

Major comments.

  • The authors employed MDCK as a control cell line for the in vitro experiments; however, MDCK is an epithelial cell line and does not possess relevance to B cells. It is advisable to utilize non-neoplastic canine lymphocytes (PBMCs) as an appropriate control.
  • Most of the genes are primarily measured by RT-PCR, which indicates mRNA levels. These transcriptional differences could result from negative feedback mechanisms, so it would be helpful to examine the genes at the protein level using a Western blot.

Author Response

For research article

Response to Reviewer 3 Comments

1. Summary

We would like to thank you for agreeing to review this paper and for the suggestions and comments that for sure will improve our paper. We hope we have addressed the various questions and queries in the revised manuscript. Here, we describe the changes that we have made regarding the comments, changes are highlighted in blue. Please find detailed responses below.

2. Questions for General Evaluation

Reviewer’s Evaluation

Response and Revisions

Does the introduction provide sufficient background and include all relevant references?

Can be improved

Are all figures and tables clear and well-presented?

Can be improved

Is the research design appropriate?

Must be improved

Are the methods adequately described?

Must be improved

Are the results clearly presented?

Can be improved

Are the conclusions supported by the results?

Can be improved

3. Point-by-point response to Comments and Suggestions for Authors

Comments 1: The authors employed MDCK as a control cell line for the in vitro experiments; however, MDCK is an epithelial cell line and does not possess relevance to B cells. It is advisable to utilize non-neoplastic canine lymphocytes (PBMCs) as an appropriate control

Response 1: We agreed with the observation about MDCK may not be the ideal control to be compared to B cells. Unfortunately, there are no established cell lines of normal lymphoid origin derived from dogs, leaving us as you suggest with the possibility of using PBMCs as control. The problem is that the use of PBMCs taken from healthy donors is that it would inevitably be a heterogeneous population of different cell types, increasing variability in results. As our objective was to study the anticancer properties of G4 in canine cells, we only needed a non-cancerous cell as control to identify differences when treated with PhenDC3 when comparing non-cancer and cancer cells. We decided to go for the MDCK cell line as it is a very well-known and used cell line in veterinary science.

Comments 2: Most of the genes are primarily measured by RT-PCR, which indicates mRNA levels. These transcriptional differences could result from negative feedback mechanisms, so it would be helpful to examine the genes at the protein level using a Western blot

Response 2: Adding the information about protein expression is an interesting point and we highly appreciate the suggestion. We have included some preliminary blots of Chk1 and Rad51 proteins, and indeed, we have observed some interesting results in the case of CLB70. However, to really identify the reason for those transcriptional differences many more experiments will be needed. We decided to include the preliminary data on the supplementary material and briefly discuss it in the discussion section. Changes are highlighted in light blue.

4. Response to Comments on the Quality of English Language

Point 1: The English is fine and does not require any improvement

Response 1: NA

5. Additional clarifications

There is nothing else to add.

Round 2

Reviewer 3 Report

Comments and Suggestions for Authors

The authors responded to my comments. The WB data appears interesting, and I hope they explore this further in their future studies.